# Longitudinal association between handgrip strength, gait speed and risk of serious falls in a community-dwelling older population

Thao Pham [1], John J. McNeil[1], Anna L. Barker[1], Suzanne G. Orchard[1], Anne B. Newman[2], Catherine Robb[1], Michael E. Ernst[3,4], Sara Espinoza[5,6], Robyn L. Woods[1], Mark R. Nelson[7], Lawrence Beilin[8], Sultana Monira Hussain [1,9]*

1 School of Public Health and Preventive Medicine, Monash University, Melbourne, Victoria, Australia, 2 Center for Aging and Population Health, Department of Epidemiology, University of Pittsburgh, Pittsburgh, PA, United States of America, 3 Department of Pharmacy Practice and Science, College of Pharmacy, The University of Iowa, Iowa City, Iowa, United States of America, 4 Department of Family Medicine, Carver College of Medicine, The University of Iowa, Iowa City, Iowa, United States of America, 5 Division of Geriatrics, Gerontology & Palliative Medicine, Sam and Ann Barshop Institute for Longevity and Aging Studies, UT Health San Antonio, San Antonio, Texas, United States of America, 6 Geriatrics Research, Education and Clinical Center, South Texas Veterans Health Care System, San Antonio, Texas, United States of America, 7 Menzies Institute for Medical Research, University of Tasmania, Hobart, Tasmania, Australia, 8 Medical School, Royal Perth Hospital, University of Western Australia, Perth, Australia, 9 Department of Medical Education, Melbourne Medical School, The University of Melbourne, Melbourne, Victoria, Australia

* Monira.hussain@monash.edu

**Data Availability Statement:** Data contain potentially sensitive patient information. However, data request may be sent to Monash University

## Abstract

### Objective

Both grip strength and gait speed can be used as markers of muscle function, however, no previous study has examined them in the same population with respect to risk of falls.

### Methods

In this prospective cohort study, utilising data from the ASPirin in Reducing Events in the Elderly (ASPREE) trial and ASPREE-Fracture substudy, we analysed the association of grip strength and gait speed and serious falls in healthy older adults. Grip strength was measured using a handheld dynamometer and gait speed from 3-metre timed walks. Serious falls were confined to those involving hospital presentation. Cox regression was used to calculate hazard ratios (HR) and 95% confidence intervals (CI) for associations with falls.

### Results

Over an average of 4.0±1.3 years, amongst 16,445 participants, 1,533 had at least one serious fall. After adjustment for age, sex, physical activity, body mass index, Short Form 12 (state of health), chronic kidney disease, polypharmacy and aspirin, each standard deviation (SD) lower grip strength was associated with 27% (HR 1.27, 95% CI 1.17–1.38) higher risk of falls. The results remained the same for males and females. There was a dose-response relationship in the association between grip strength and falls risk. The higher risk of falls was observed in males in all body mass index (BMI) categories, but only in obese females.

using the following email address aspree.
ams@monash.edu.

**Funding:** Source of Funding: The ASPirin in
Reducing Events in the Elderly (ASPREE) study
was supported by grants from the National
Institute on Aging and the National Cancer Institute
at the National Institutes of Health
(U01AG029824); the National Health and Medical
Research Council of Australia (334037 and
1127060); Monash University (Melbourne, VIC,
Australia); and the Victorian Cancer Agency
(Australia). The ASPREE-Falls & Fractures sub-
study was supported by a grant from the National
Health and Medical Research Council of Australia
(1067242). Role of the Funder/Sponsor: The
sponsor had no role in the design and conduct of
this study; collection, management, analysis, and
interpretation of the data and decision to submit
the manuscript for publication but was given the
opportunity to review and comment on the
manuscript.

**Competing interests:** Dr Hussain is the recipient of
National Health and Medical Research Council
(NHMRC) Early Career Fellowship (APP1142198),
Professor McNeil is supported through an NHMRC
Leadership Fellowship (IG 1173690). No other
disclosures are reported by the other authors.

The association between gait speed and falls risk was weaker than the association between
grip strength and falls risk.

## Conclusions

All males and only obese females with low grip strength appear to be at the greatest risk of
serious falls. These findings may assist in early identification of falls.

## Introduction

Falls are associated with reduced independence and are the second leading cause of death due
to injury among people over 60 years of age [1]. In the United States, approximately $50 billion
is spent on medical costs related to non-fatal fall injuries and $754 million on fatal falls each
year [2]. In Australia, among those aged 65 years and over in 2019–20, falls resulted in over
133,000 hospitalisations and 5,000 deaths [3]. The average health system cost per fall injury in
the Republic of Finland and Australia were USD3611 and USD1049, respectively [4].

Several factors can increase falls risk such as mobility impairment, loss of balance [5], older
age, being female, frailty, living with chronic conditions (e.g., cardiovascular disease, diabetes,
osteoarthritis), as well as medications used to treat these conditions, including polypharmacy
[6]. Among these risk factors, identification of modifiable factors which influence balance
such as muscle strength [7] might be important for preventing falls in older adults. Thus, a bet-
ter understanding of the influences of muscle strength on falls is crucial in planning interven-
tions to prevent falls among older adults.

In many studies, muscle strength has been measured using hand grip strength [8]. Similarly,
gait speed is increasingly recognised as a measure of lower limb muscle function [5]. Previous stud-
ies have examined the association between grip strength and falls risk and reported inconclusive
findings [9–11]. A previously published systematic review including five publications and a recent
cross-sectional study showed that low gait speed is associated with increased falls risk [6, 12]. The
MOBILIZE Boston Study showed that faster gait speed was associated with an increased risk of
outdoor falls and slower gait speed was associated with an increased risk of indoor falls [13]. The
reason for these inconclusive findings for both the associations between grip strength and falls [9–
11] and gait speed and falls [6, 12, 13] may be due to the small sample size of these studies, differ-
ences in the study populations examined with respect to existing chronic disease and the data on
falls were mostly self-reported. All studies included in the systematic review except the MOBILIZE
Boston Study, examining the associations between gait speed and falls were not adjusted for con-
founders or only adjusted for age and sex [6, 12, 13]. Thus, it is still unclear whether grip strength
and gait speed are important independent predictors of falls irrespective of distinct subclinical
chronic diseases, medication use, etc. No previous study has examined the association between
grip strength and gait speed in the same population, except the Osteoporotic Fractures in Men
(MrOS) Study [14]. In this current study, we examined the association of hand grip strength and
gait speed and serious fall-related hospital presentation in a large community-based apparently
healthy older adult population, free from significant age-related diseases at baseline.

## Materials and methods

### Participants, design and setting

This study used data from the ASPirin in Reducing Events in the Elderly (ASPREE) and
ASPREE-Fracture substudy which collected detailed information on serious falls from

Australian participants. Ethics approvals were obtained from the Monash University Human Research Ethics Committee (MUHREC) for both the ASPREE principal trial and the ASPREE-Fracture substudy. Written informed consent was obtained from all participants.

ASPREE was a double-blinded, randomised, placebo-controlled primary prevention trial of aspirin that recruited 19,114 community-based older adults, between March 2010 and December 2014. ASPREE participants were free from cardiovascular disease (CVD) events, dementia, physical disability, or chronic illness expected to limit survival to less than 5 years. A total of 16,703 participants (aged ≥70 years) were recruited within Australia, and 2,411 participants (aged ≥65 years) were recruited within the United States [15, 16]. Further details regarding screening, recruitment and trial design have been published previously [16].

## Hospital presentation for falls

The ASPREE-Fracture sub-study collected data on all fractures or hospital presentations resulting from a serious fall that occurred post-randomisation. A serious fall was defined as an event which resulted in a person coming to rest inadvertently on the ground or floor or *other* lower level and resulted in presentation to a hospital [17]. During annual in-person visits and 6-month telephone follow-up, participants were asked about hospital presentations that had occurred because of a serious fall or fracture(s) within the previous six months. All potential fractures or serious fall-related hospital presentation events were followed up by data collection from general practice, specialist, and hospital medical records. Verifying serious fall-related hospital admission was undertaken by reviewing these records and confirming the fall event date. All potential serious falls were reviewed by an endpoint adjudication committee consisting of clinicians and research personnel blinded to ASPREE treatment group allocation [18].

## Demographic data and anthropometric measurements

The ASPREE principal trial collected data from in-person visits, 6-monthly telephone contacts, reviews of general practice and hospital records, and death certificates. Anthropometric data including, body mass index (BMI), physical activity (walking less than 15 minutes outside) and laboratory measurements, medical morbidities, lifestyle and socio-demographic factors, prescription medications and other health-related parameters. Modified Fried frailty phenotype, which defines frailty as the presence of weakness, slowness, exhaustion, low physical activity and weight loss, was used to categorise frailty status. Anyone with one or two criteria was categorised as prefrail, and three or more criteria were categorised as frail [19, 20].

## Grip strength and gait speed

During recruitment, grip strength (kilogram force (kgf)) was measured using a handheld dynamometer in a seated position with the arm rested at 90˚. A maximum of 3 measurements on each hand with a 15–20 second rest in between was recorded. The average grip strength of the dominant hand was used in this analysis. Gait speed (metre per second (m/s)) was measured from participants completing two timed walks of 3 metres at usual pace from standing start, with at least 1 metre at the end of the course to prevent slowing. The mean average of two walks was used in this analysis.

## Statistical analysis

The distribution of hand grip strength showed differences between males and females (S1 Fig in S1 File). Grip strength was subsequently categorised according to sex-specific quintiles of standard deviation (SD). The categories were low grip strength (quintile 1, lowest 20%),

medium grip strength (quintile 2–4, middle 60%) and high grip strength (quintile 5, highest 20%).

Initially, grip strength was treated as a continuous variable. Following that, it was treated as a categorical variable, with high grip strength as the reference category. Furthermore, the European Working Group on Sarcopenia in Older People (EWGSOP) [21] was used to examine the association between grip strength and falls. The distribution of gait speed for male and female were similar (S2 Fig in S1 File), gait speed was categorised as low gait speed (quintile 1, lowest 20%), medium gait speed (quintile 2–4, middle 60%) and high gait speed (quintile 5, highest 20%). The reference category for gait speed was high gait speed. We have checked the correlation between grip strength and gait speed by using Pearson-correlation test.

Baseline general characteristics of participants were analysed according to grip strength and gait speed categories. Analysis of variance (ANOVA) was used for continuous variables and $x^2$ tests for categorical variables. Cox proportional hazards regression was used to calculate hazard ratio (HR) and 95% CI for hand grip strength from the time of randomisation to the first serious fall-related hospital presentation.

Analyses of grip strength and risk of falls were adjusted for *a priori* selected age, sex (Model 1); Model 1 and physical activity, BMI and Short Form 12 (state of health) (Model 2); Model 2 and chronic kidney disease and polypharmacy (Model 3); Model 3 and aspirin (100 mg) (Model 4). Analyses for gait speed and risk of falls were adjusted with the same covariate adjustments as grip strength, with an additional variable for walking aid in all analysis models.

We examined if age and sex or BMI and sex modified the association between grip strength and falls by examining the interaction. There was an interaction between BMI and sex (p = 0.06), and hence, stratified analyses according to BMI categories and sex were also performed in sensitivity analyses. Additionally, for grip strength analyses, we excluded frail and prefrail participants and repeated the analyses since prefrailty/frailty is associated with several adverse effects that increases the risk of falls [22]. Similarly, for gait speed, we repeated the analyses excluding those who use a walking aid.

Statistical analyses were performed using Stata MP version 17 for Windows (StataCorp LP, College Station, TX).

## Results

A total of 16,445 participants were included in the grip strength analysis, including 1,512 participants that experienced at least one serious fall. A total of 16,616 participants were included in the gait speed analysis, including 1,533 participants that experienced at least one serious fall (S3 Fig in S1 File). Participants were followed up over an average of 4.0 (SD 1.3) years. The overall incidence rate of falls in our participants was 23 per 1000 person-years. There was no/mild correlation between grip strength and gait speed (Pearson-correlation coefficient = 0.25, p <0.001).

The baseline characteristics of the participants according to grip strength categories are shown in Table 1. Participants with low grip strength were less likely to engage in physical activity, more often have chronic kidney disease and more likely to be treated with multiple medications (polypharmacy) compared to those with higher grip strength. 5.3% serious falls were observed in people with high grip strength, whereas 13.7% serious falls were observed in those who had a low grip strength (Fig 1). The rate of serious falls was higher in males with a grip strength <30 kgf and female with a grip strength <20 kgf (Fig 2).

The baseline characteristics of participants according to gait speed categories are shown in S1 Table in S1 File. Participants with low gait speed were less likely to engage in physical activity, have chronic kidney disease and treated with polypharmacy compared to those with high gait speed.

**Table 1. Baseline characteristics of the included participants overall and according to categories of grip strength.**

|  | Overall | High (Q5) | Medium (Q2-Q4) | Low (Q1) | p-value |
|---|---|---|---|---|---|
| **N (%)** | 16445 (100) | 3126 (19.01) | 9957 (60.55) | 3362 (20.44) | |
| **Age in years** | 75.29 (4.36) | 73.5 (3.07) | 75.17 (4.14) | 77.31 (5.13) | <0.001 |
| **Females, n (%)** | 9015 (54.82) | 1676 (53.61) | 5495 (55.19) | 1844 (54.85) | 0.31 |
| **Low activity, n (%)** | 1049 (6.38) | 145 (4.64) | 609 (6.12) | 295 (8.77) | <0.001 |
| **BMI (kg/m²)** | 28.00 (4.60) | 28.19 (4.34) | 28.00 (4.60) | 27.89 (4.84) | <0.001 |
| **Waist circumference (cm)** | 97.10 (12.68) | 97.59 (12.48) | 96.87 (12.61) | 97.31 (13.08) | 0.013 |
| **Smoking history, n (%)** | | | | | 0.032 |
| Current/Former | 7307 (44.43) | 1419 (45.39) | 4460 (44.79) | 1428 (42.47) | |
| Never | 9138 (55.57) | 1707 (54.61) | 5497 (55.21) | 1934 (57.53) | |
| **Alcohol use, n (%)** | | | | | <0.001 |
| Current/Former | 13786 (83.83) | 2679 (85.70) | 8363 (83.99) | 2744 (81.62) | |
| Never | 2659 (16.17) | 447 (14.30) | 1594 (16.01) | 618 (18.38) | |
| **SF12 (state of health),** n (%) | | | | | <0.001 |
| Good health | 15758 (95.86) | 3048 (97.50) | 9575 (96.20) | 3135 (93.30) | |
| Fair/Poor health | 681 (4.14) | 78 (2.50) | 378 (3.80) | 225 (6.70) | |
| **Systolic blood pressure** | 139.80 (16.33) | 140.06 (16.15) | 139.85 (16.24) | 139.41 (16.74) | 0.063 |
| **Chronic kidney disease, n (%)** | 3992 (26.28) | 702 (24.32) | 2316 (25.24) | 974 (31.14) | <0.001 |
| **Polypharmacy#, n (%)** | 4274 (26.0) | 604 (19.3) | 2536 (25.5) | 1134 (33.7) | <0.001 |
| **On trial medication (Aspirin, 100mg), n (%)** | 8202 (49.88) | 1522 (48.69) | 4985 (50.07) | 1695 (50.42) | 0.317 |
| **Exposure** | | | | | |
| Grip strength (kgf) (median and IQR), males | 35 (8.7, 62) | 35.3 (33.3, 37) | 31 (28.67, 33) | 24.7 (8.7, 28.3) | |
| Grip strength (kgf) (median and IQR), females | 20.7 (0, 40.7) | 20.7 (19.7, 22.0) | 18.0 (16.7, 19.3) | 14.0 (0, 16.3) | |
| **Outcome** | | | | | |
| Falls n, (%) | 1512 (9.20) | 164 (5.25) | 890 (8.94) | 459 (13.65) | <0.001 |
| Incidence rate per 1000 person-year (95% CI) | 23 (22, 24) | 34 (31, 37) | 23 (21, 24) | 13 (11, 15) | |

Kgf = kilogram force; BMI = body mass index; SF12 = Short Form 12; IQR = interquartile range

*Grip strength (kgf) categories: Low (Q1, lowest 20%), Medium (Q2-Q4, middle 60%), High (Q5, highest 20%)

**SF12 (state of health): measure of self-reported health status

***Chronic kidney disease baseline: eGFR > an estimated glomerular filtration rate of less than 60 ml per minute per 1.73 m2 or a ratio of albumin (in milligrams per litre) to creatinine (in millimoles per litre) in urine of 3 or more

#Polypharmacy: the use of >5 medications

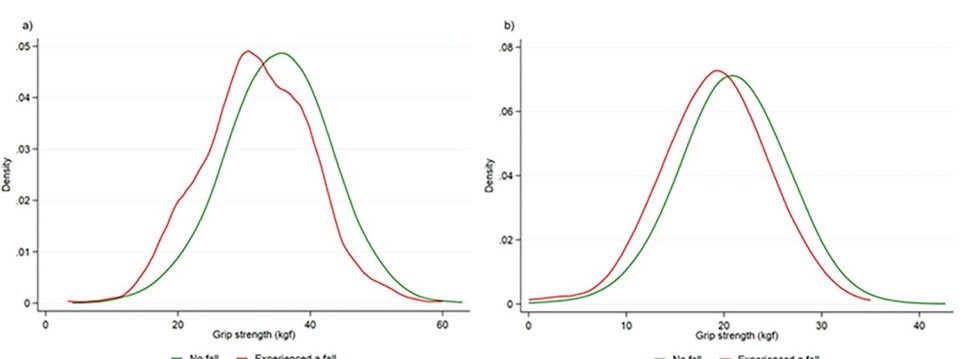

**Fig 1. Distribution of grip strength (kilogram force) according to no fall vs experiencing a serious fall.** a) Male, b) Female.

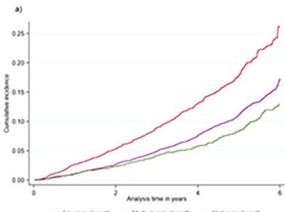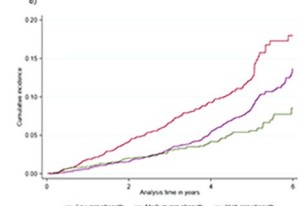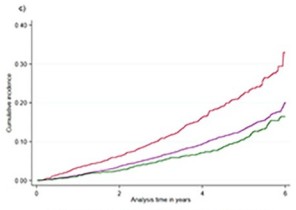

**Fig 2. Nelson-Aalen cumulative hazard estimates–incidence of serious falls according to grip strength (kilogram force) categories.** a) All participant, b) Male, c) Female.

## Associations between grip strength and risk of serious falls

Table 2 presents the association between grip strength and serious falls. In the fully adjusted model, each SD lower grip strength was associated with 27% (HR 1.27, 95% CI 1.17, 1.38) increase in risk of serious falls for all participants. In sex-stratified analysis, the results remained similar for males (HR 1.28, 95% CI 1.14, 1.44) and females (HR 1.27, 95% CI 1.12, 1.43). When risk of serious falls amongst those with high grip strength (quintile 5) was compared with the medium and low grip strength groups, the risk of serious falls increased by 47% (HR 1.47, 95% CI 1.24, 1.76) for all participants with medium grip strength (quintile 2–4) and 73% (HR 1.73, 95% CI 1.42, 2.10) for those with low grip strength (quintile 1) in the fully

**Table 2. Association between grip strength and risk of falls (HR, 95% CI).**

| | Model 1 | Model 2 | Model 3 | Model 4 |
|---|---|---|---|---|
| **All Population (n = 16 445)** | | | | |
| **Linear association e(ach SD decrease)** | 1.30 (1.22, 1.43) | 1.30 (1.20, 1.41) | 1.27 (1.17, 1.38) | 1.27 (1.17, 1.38) |
| **Grip strength (kgf) quintiles*** | | | | |
| **High (Q5)** | Ref | Ref | Ref | Ref |
| **Medium (Q2-Q4)** | 1.44 (1.22, 1.71) | 1.44 (1.21, 1.70) | 1.48 (1.24, 1.76) | 1.47 (1.24, 1.76) |
| **Low (Q1)** | 1.79 (1.49, 2.15) | 1.75 (1.45, 2.11) | 1.73 (1.42, 2.10) | 1.73 (1.42, 2.10) |
| **Stratified by sex** | | | | |
| **Males** ** **(n = 7430)** | | | | |
| **Linear association (each SD decrease)** | 1.32 (1.18, 1.48) | 1.30 (1.17, 1.46) | 1.28 (1.14, 1.44) | 1.28 (1.14, 1.44) |
| **Grip strength (kgf) quintiles** | | | | |
| **High (Q5)** | Ref | Ref | Ref | Ref |
| **Medium (Q2-Q4)** | 1.76 (1.29, 2.40) | 1.76 (1.29, 2.40) | 1.94 (1.39, 2.71) | 1.94 (1.39, 2.71) |
| **Low (Q1)** | 2.27 (1.62, 3.18) | 2.21 (1.57, 3.11) | 2.30 (1.60, 3.32) | 2.30 (1.60, 3.32) |
| **Females** ** **(n = 9015)** | | | | |
| **Linear association (each SD decrease)** | 1.32 (1.18, 1.48) | 1.30 (1.16, 1.46) | 1.27 (1.43, 1.13,) | 1.27 (1.12, 1.43) |
| **Grip strength (kgf) quintiles** | | | | |
| **High (Q5)** | Ref | Ref | Ref | Ref |
| **Medium (Q2-Q4)** | 1.31 (1.08, 1.61) | 1.31 (1.07, 1.60) | 1.31 (1.07, 1.62) | 1.31 (1.07, 1.61) |
| **Low (Q1)** | 1.61 (1.29, 2.01) | 1.57 (1.26, 1.97) | 1.53 (1.21, 1.93) | 1.52 (1.21, 1.92) |

Kgf = kilogram force; SD = standard deviation; Ref = reference.

Data presented as hazard ratio [HR, 95% confidence interval (CI)]. Model 1: age and gender. Model 2: age, gender, physical activity, BMI and self-reported health status. Model 3: age, gender, physical activity, BMI, SF12 (state of health), chronic kidney disease, and polypharmacy. Model 4: age, gender, physical activity, BMI, SF12 (state of health), chronic kidney disease, polypharmacy and aspirin (100mg).

*Grip strength (kgf) categories: Low (Q1, lowest 20%), Medium (Q2-Q4, middle 60%), High (Q5, highest 20%)

**Not adjusted for gender

adjusted model. Similarly, lower grip strength increased the risk of serious falls when these analyses were stratified by sex. For males, compared with the high grip strength category (quintile 5), there was a 94% increased risk in the medium grip strength category (HR 1.94, 95% CI 1.39, 2.71) and a 2.3-fold increased risk in the low grip strength category (HR 2.30, 95% CI 1.60, 3.32). For females, there was a 31% increased risk in the medium grip strength category (HR 1.31, 95% CI 1.07, 1.61) and a 52% increased risk in the low grip strength category (HR 1.52, 95% CI 1.21, 1.92). Using the EWGSOP cut-off points for grip strength, we found similar results (S2 Table in S1 File).

## Associations between gait speed and risk of serious falls

Table 3 presents the association between gait speed and the risk of serious falls. In the adjusted models, one SD decrease in gait speed was not associated with the risk of serious falls. However, when examined as gait speed categories, we found those in the lowest quintile of gait speed had a 48% increased risk of serious falls (HR 1.48, 95% CI 1.24, 1.77) compared with the highest quintile. Similar results were observed when males and females were compared separately i.e., males (HR 1.53, 95% CI 1.13, 2.09), females (HR 1.46, 95% CI 1.18, 1.82).

## Sensitivity analyses

The association between grip strength and the risk of serious falls according to BMI categories are presented in S4 Table in S1 File. Amongst overweight males, one SD lower grip strength

**Table 3. Association between gait speed and risk of falls (HR, 95% CI).**

| | Model 1 | Model 2 | Model 3 | Model 4 |
|---|---|---|---|---|
| **All Population** (n = 16 445) | | | | |
| **Linear association (each SD decrease)** | 0.998 (0.997, 0.998) | 0.998 (0.997, 0.998) | 0.998 (0.997, 0.999) | 0.998 (0.997, 0.999) |
| **Gait speed (m/s) quintiles*** | | | | |
| High (Q5) | Ref | Ref | Ref | Ref |
| Medium (Q2-Q4) | 1.16 (1.00, 1.34) | 1.15 (1.00, 1.34) | 1.18 (1.01, 1.37) | 1.18 (1.01, 1.37) |
| Low Q1 | 1.55 (1.31, 1.83) | 1.50 (1.27, 1.78) | 1.48 (1.24, 1.77) | 1.48 (1.24, 1.77) |
| **Stratified by sex** | | | | |
| **Males** ** (n = 7430) | | | | |
| **Linear association (each SD decrease)** | 0.997 (0.995, 0.998) | 0.997 (0.995, 0.999) | 0.997 (0.996, 0.999) | 0.997 (0.996, 0.999) |
| **Gait speed (m/s) quintiles*** | | | | |
| High (Q5) | Ref | Ref | Ref | Ref |
| Medium (Q2-Q4) | 1.27 (0.98, 1.65) | 1.23 (0.95, 1.60) | 1.26 (0.96, 1.65) | 1.26 (0.96, 1.65) |
| Low (Q1) | 1.71 (1.28, 2.29) | 1.56 (1.16, 2.10) | 1.53 (1.13, 2.09) | 1.53 (1.13, 2.09) |
| **Females** ** (n = 9015) | | | | |
| **Linear association (each SD decrease)** | 0.998 (0.997, 0.999) | 0.998 (0.997, 0.999) | 0.998 (0.997, 1.000) | 0.998 (0.997, 1.000) |
| **Gait speed (m/s) quintile*** | | | | |
| High (Q5) | Ref | Ref | Ref | Ref |
| Medium (Q2-Q4) | 1.10 (0.92, 1.32) | 1.12 (0.94, 1.34) | 1.14 (0.95, 1.37) | 1.14 (0.95, 1.37) |
| Low (Q1) | 1.47 (1.20, 1.81) | 1.48 (1.20, 1.83) | 1.46 (1.18, 1.82) | 1.46 (1.18, 1.82) |

m/s = metre per second; SD = standard deviation; Ref = reference.

Data presented as hazard ratio [HR, 95% confidence interval (CI)]. Model 1: age, gender, and walking aid. Model 2: age, gender, walking aid, physical activity, BMI and SF12 (state of health). Model 3: age, gender, walking aid, physical activity, BMI, SF12 (state of health), chronic kidney disease, and polypharmacy. Model 4: age, gender, walking aid, physical activity, BMI, SF12 (state of health), chronic kidney disease, polypharmacy and aspirin (100mg).

*Gait speed (m/s) categories: Low (Q1, lowest 20%), Medium (Q2-Q4, middle 60%), High (Q5, highest 20%)

**Not adjusted for gender

increased the risk of serious falls by 33% (HR 1.33, 95% CI 1.11, 1.58). Similarly, in obese males, one SD lower grip strength increased the risk of serious falls by 35% (HR 1.35, 95% CI 1.10, 1.65). When grip strength was analysed as categorical variables, high risk of serious falls was observed in those with medium and low grip strength amongst overweight and obese males. In contrast, in normal weight and overweight females, lower grip strength was not associated with serious falls risk. Only obese females with a lower hand grip strength had an increased risk of serious falls. For example, one SD lower grip strength increased the risk of serious falls by 47% (HR 1.47, 95% CI 1.19, 1.82). When grip strength was analysed as a categorical variable, those with low grip strength had an increased risk of serious falls by 94% (HR 1.94, 95% CI 1.26, 2.98) compared to those with high grip strength.

The associations between grip strength and serious falls persisted when we excluded the participants who were prefrail/frail (n = 6300) (S3 Table in S1 File). The associations between gait speed and risk of serious falls persisted when excluding participants who use a walking aid (n = 364) (S5 Table in S1 File).

## Discussion

The principal finding of this study was that a lower hand grip strength in older adults was associated with a substantially increased risk of serious falls, resulting in hospital presentation. There was a dose-response relationship in the association between grip strength and risk of serious falls such that the lower the grip strength, the higher the risk of serious falls. When the analyses were stratified by BMI categories, the risk of serious falls was increased in males irrespective of BMI categories. However, the increased risk of serious falls in females was mainly confined in the obese BMI category. A low gait speed was also predictive of serious falls but with a weaker relationship than grip strength. Furthermore, the association between gait speed and serious falls was confined only among participants who were in the lowest gait speed category, but not in the middle gait speed category.

Our finding that decreasing hand grip strength is a risk factor for serious falls are in agreement with studies previously published [9, 10, 23–27]. However, most of the previously published studies were cross-sectional [23–27] except for the Tasmanian Older Adult Cohort (TASOAC) study [9] and a cohort study of Brazilian women aged 60 years and over [10]. There are several differences between ASPREE and TASOAC. For example, TASOAC participants were younger (51.1–79.9 years of age), and pre-existing illnesses were not excluded. In addition, falls risk at 10 years was assessed using the Short Form Physiological Profile Assessment [9], actual falls were not recorded. The analyses were adjusted for age only [9] but several risk factors are associated with falls [6]. Similarly, the Brazilian study had several limitations as falls were self-reported and the participants were not free from subclinical diseases [10]. Our finding was inconsistent with the Korean Longitudinal Study of Aging (KLoSA), where grip strength was not associated with the risk of falls [11]. Perhaps in this study, participants did not show variations in grip strength, since 39% of the study participants had cognitive decline/ dementia, and 18.5% participants had physical disability. The number of participants with frailty and prefrailty was not mentioned [11]. This study also used self-reported fall events [11].

Our study extends beyond the TASOAC and the Brazilian women's study as we included a larger population (TASOAC vs Brazilian women's study vs ASPREE: 1,041 vs 195 vs 16,445), including both sexes, initially relatively healthy population and adjusted for several risk factors: age, gender, physical activity, BMI, Short Form 12 (state of health), chronic kidney disease, polypharmacy and aspirin (100mg). The TASOAC study also failed to identify the difference in grip strength between males and females. In contrast, the ASPREE females had lower grip

strength than males, which is supported by other studies that females have lower average hand grip strength than males [28].

In stratified analyses by BMI category, we found that the risk of serious falls increased in males irrespective of BMI category. In females, the risk of falls was confined mainly to those who were obese. These analyses demonstrate a new finding and may be explained by the different characteristics of body composition and grip strength in males and females [29]. Sarcopenic obesity is more common in older females than males [30]. Furthermore, on average, grip strength for males decreases faster with age than it does for females [27]. The gender difference tends to narrow slightly with older age [31].

A potential explanation of these findings was that low grip strength is an indicator of frailty which, in turn, is a risk factor for serious falls amongst ageing individuals [32]. However, in sensitivity analyses, we have excluded participants who were prefrail or frail and found a similar relationship. This finding suggests that grip strength reflects muscle strength more generally which in turn is a major determinant of serious falls risk within this population. Low muscle strength has been shown to be associated with worse dynamic balance, higher fear of falling, and the occurrence of falls over the previous year [33], which are predictors of future falls [4, 34, 35].

We found a weaker association between lower gait speed and serious falls than with lower grip strength and serious falls. Whereas previous studies showed that low gait speed is common in people who experienced a fall or it increases incidence of falls [6, 12]. These studies have methodological differences than the current study in regards to sample size [6, 12, 13], confounding adjustment [6, 12], outcome measurement [6, 12] etc. The contrasting findings from these studies can be due to differences in how gait speed was measured. In the previous studies, participants were required to walk 8–10 meters for gait speed measurement [6, 12, 13]. Whereas in this current study, participants were required to walk 3 meters. Another possibility could be slower walkers are more likely to be less active and/or cautious during daily activities.

The strength of our study includes its large sample size, intensive data collection from in-person visits, telephone assessments, rigorous collection of relevant ancillary information, and thorough objective assessment of hand grip strength and gait speed. Falls were verified by reviewing clinical records that were then adjudicated to ensure robust case ascertainment. Our results need to be considered within the study's limitations. This analysis is a post-hoc analysis of data collected as part of a clinical trial; thus, its findings may have arisen by chance, although the internal consistency of the findings makes this less likely. Lastly, the definition of serious falls required a hospital presentation, which may have underestimated the overall number of falls.

## Conclusion

The clinical implications of these findings are that the measurement of hand grip strength can provide a simple and objective way of identifying older adults with a greater likelihood of having a serious fall. Based on this study, all males and obese females with low grip strength appear to be at the greatest risk for serious falls. The early identification of these most susceptible individuals would enable targeted prevention strategies, that may include muscle strengthening through resistance exercise [36], to reduce risk of serious falls in older populations.

## Supporting information

**S1 File.**
(DOCX)

## Acknowledgments

We thank the patients who participated in this trial.

## Author Contributions

**Conceptualization:** John J. McNeil, Sultana Monira Hussain.

**Data curation:** John J. McNeil, Anna L. Barker, Suzanne G. Orchard, Anne B. Newman, Robyn L. Woods, Mark R. Nelson, Lawrence Beilin.

**Formal analysis:** Thao Pham, Sultana Monira Hussain.

**Funding acquisition:** John J. McNeil, Anna L. Barker, Robyn L. Woods, Sultana Monira Hussain.

**Investigation:** Catherine Robb, Mark R. Nelson, Sultana Monira Hussain.

**Methodology:** Thao Pham, John J. McNeil, Anne B. Newman, Michael E. Ernst, Robyn L. Woods, Lawrence Beilin, Sultana Monira Hussain.

**Project administration:** John J. McNeil, Robyn L. Woods, Sultana Monira Hussain.

**Software:** John J. McNeil.

**Supervision:** Sultana Monira Hussain.

**Validation:** Michael E. Ernst, Sara Espinoza, Sultana Monira Hussain.

**Visualization:** Sultana Monira Hussain.

**Writing – original draft:** Thao Pham.

**Writing – review & editing:** Thao Pham, John J. McNeil, Anna L. Barker, Suzanne G. Orchard, Anne B. Newman, Catherine Robb, Michael E. Ernst, Sara Espinoza, Robyn L. Woods, Mark R. Nelson, Lawrence Beilin, Sultana Monira Hussain.

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
