## [Decision Letter · Decision Letter 0]

31 Jan 2023

PONE-D-23-00361Longitudinal association between handgrip strength, gait speed and risk of serious falls in a community-dwelling older populationPLOS ONE

Dear Dr. Hussain,

Thank you for submitting your manuscript to PLOS ONE. After careful consideration, we feel that it has merit but does not fully meet PLOS ONE’s publication criteria as it currently stands. Therefore, we invite you to submit a revised version of the manuscript that addresses the points raised during the review process.

We look forward to receiving your revised manuscript.

Kind regards,

Mario Ulises Pérez-Zepeda, M.D., Ph.D.

Academic Editor

PLOS ONE

“Source of Funding: The ASPirin in Reducing Events in the Elderly (ASPREE) study was supported by grants from the National Institute on Aging and the National Cancer Institute at the National Institutes of Health (U01AG029824); the National Health and Medical Research Council of Australia (334037 and 1127060); Monash University (Melbourne, VIC, Australia); and the Victorian Cancer Agency (Australia).  The ASPREE-Falls & Fractures sub-study was supported by a grant from the National Health and Medical Research Council of Australia (1067242).

Role of the Funder/Sponsor: The sponsor had no role in the design and conduct of this study; collection, management, analysis, and interpretation of the data and decision to submit the manuscript for publication but was given the opportunity to review and comment on the manuscript.

Additional Contributions: We thank the patients who participated in this trial.

Conflict of Interest

Dr Hussain is the recipient of National Health and Medical Research Council (NHMRC) Early Career Fellowship (APP1142198), Professor McNeil is supported through an NHMRC Leadership Fellowship (IG 1173690). No other disclosures are reported by the other authors."

‘Source of Funding: The ASPirin in Reducing Events in the Elderly (ASPREE) study was supported by grants from the National Institute on Aging and the National Cancer Institute at the National Institutes of Health (U01AG029824); the National Health and Medical Research Council of Australia (334037 and 1127060); Monash University (Melbourne, VIC, Australia); and the Victorian Cancer Agency (Australia).  The ASPREE-Falls & Fractures sub-study was supported by a grant from the National Health and Medical Research Council of Australia (1067242).

Role of the Funder/Sponsor: The sponsor had no role in the design and conduct of this study; collection, management, analysis, and interpretation of the data and decision to submit the manuscript for publication but was given the opportunity to review and comment on the manuscript.

Conflict of Interest

Dr Hussain is the recipient of National Health and Medical Research Council (NHMRC) Early Career Fellowship (APP1142198), Professor McNeil is supported through an NHMRC Leadership Fellowship (IG 1173690). No other disclosures are reported by the other authors.”

“No conflict of interests was declared by any of the authors.”

Reviewers' comments:

Reviewer's Responses to Questions

**Comments to the Author**

1. Is the manuscript technically sound, and do the data support the conclusions?

Reviewer #1: Yes

Reviewer #2: Partly

2. Has the statistical analysis been performed appropriately and rigorously? 

Reviewer #1: Yes

Reviewer #2: No

3. Have the authors made all data underlying the findings in their manuscript fully available?

Reviewer #1: Yes

Reviewer #2: Yes

4. Is the manuscript presented in an intelligible fashion and written in standard English?

Reviewer #1: Yes

Reviewer #2: Yes

5. Review Comments to the Author

Reviewer #1: In this prospective cohort study, utilising data from the ASPirin in Reducing Events in the Elderly (ASPREE) trial and ASPREE-Fracture substudy, the authors analysed the association of grip strength and gait speed and serious falls in healthy older adults. They found that all males and only obese females with low grip strength appeared to be at the

greatest risk of serious falls. This may assist in early identification of falls. Limitations of this observational study are sufficiently acknowledged.

Reviewer #2: This is an interesting manuscript that evaluates the association between the handgrip strength, gait speed and risk of serious falls in a community-dwelling older population. The authors found that handgrip strength was associated with risk of falls in both genders. One SD lower of grip strength was associated with 27% higher risk of falls. The association between gait speed and risk of falls was not as significant as grip strength. However, there are some points that should be addressed in this manuscript.

1. There are absolute cu-off points for grip strength and gait speed among elderly population – it would be more interesting to evaluate the association　with normal or abnormal grip strength and gait speed on the risk of falls.

2. For Table 2 and 3 – the data were presented with gender-specification but the gender variable was still put into the models.

3. The results will be more informative if it is segregated by grip strength and gait speed – for example, there could be four matrixes with normal or abnormal grip strength or gait speed – then to evaluate the risk of falls among these four different situations .

4. Since this is a longitudinal study – the participants were measured every 6 months or one year – how about using the repeated measure analyses to evaluate the associations.

5. This study examines the associations among community-dwelling older population. However, there are more than 20% subjects with polypharmacy and multiple chronic diseases. It would be more informative if the chronic diseases and medications status could be present in the Table 1.

6. There are some studies using appendicular muscle mass rather than BMI or body fat to examine the association between sarcopenia, frailty and disability among elderly population in recently.

7. There are some important factors such as the dietary pattern, physical activity and lifestyle could be potential confounders to evaluate the associations.

6. PLOS authors have the option to publish the peer review history of their article (what does this mean?). If published, this will include your full peer review and any attached files.

Reviewer #1: No

Reviewer #2: No

---

## [Author Response · Author response to Decision Letter 0]

7 Apr 2023

Reviewers’ comments

Reviewer #1:

In this prospective cohort study, utilising data from the ASPirin in Reducing Events in the Elderly (ASPREE) trial and ASPREE-Fracture substudy, the authors analysed the association of grip strength and gait speed and serious falls in healthy older adults. They found that all males and only obese females with low grip strength appeared to be at the greatest risk of serious falls. This may assist in early identification of falls. Limitations of this observational study are sufficiently acknowledged.

Author response: We appreciate your feedback and insight into our analyses.

Reviewer #2:

1. There are absolute cut-off points for grip strength and gait speed among elderly population – it would be more interesting to evaluate the association　with normal or abnormal grip strength and gait speed on the risk of falls.

Author Response: The ASPREE study participants are initially healthy older aged >70 years. Grip strength varies according to age [1]. Furthermore, a meta-analysis on grip strength by Dodds RM et al [2] found variation in grip strength for different countries, and highlighted the need for geographical cut-off points for grip strength. Therefore, we decided to categorise grip strength and gait speed for the ASPREE population. 

However, as the reviewer suggested we compared our findings with the established cut-off points for grip strength and gait speed. The European Working Group on Sarcopenia in Older People (EWGSOP) [3] cut-off points were derived from people living in Europe. Since ASPREE participants resembles mostly with this population, we compared the grip strength and gait speed to ASPREE participants with EWGSOP. 

Grip Strength: The absolute grip strength cut-off points for EWGSOP are similar with the ASPREE categories. EWGSOP categories were: females; normal ≥16 kgf, abnormal <16 kgf and males; normal ≥27 kgf; abnormal <27 kgf. ASPREE category cut-off points were: female Q1 14.0 kgf (median), Q2 18.0 kgf, Q3 20.7 kgf and male Q1 24.7 kgf, Q2 31.0 kgf, Q3 35.3 kgf.

If we apply EWGSOP categories in ASPREE the results look similar to the results presented in the manuscript. The results are presented in the following table. 

Table: Association between grip strength and risk of falls (HR, 95% CI) using European Working Group on Sarcopenia in Older People (EWGSOP) cut-off points for grip strength

 Model 1 Model 2 Model 3 Model 4

Males (n=7430)

For all (each SD decrease) 1.32 (1.18, 1.48) 1.30 (1.17, 1.46) 1.28 (1.14, 1.44) 1.28 (1.14, 1.44)

Grip strength (kgf) as categorical variable

Normal Ref Ref Ref Ref

Low 1.45 (1.17, 1.79) 1.43 (1.15, 1.77) 1.35 (1.08, 1.69) 1.35 (1.08, 1.69)

Females (n= 9015)

For all (each SD decrease) 1.32 (1.18, 1.48) 1.30 (1.16, 1.46) 1.27 (1.43, 1.13,) 1.27 (1.12, 1.43)

Grip strength (kgf) as categorical variable

Normal Ref Ref Ref Ref

Low 1.28 (1.11, 1.48) 1.25 (1.08, 1.45) 1.21 (1.04, 1.40) 1.20 (1.04, 1.40)

Kgf = kilogram force; SD = standard deviation; Ref = reference.

Data presented as hazard ratio [HR, 95% confidence interval (CI)]. Model 1: age and gender. Model 2: age, physical activity, BMI and self-reported health status. Model 3: age, gender, physical activity, BMI, SF12 State of health, chronic kidney disease, and polypharmacy. Model 4: age, gender, physical activity, BMI, SF12 State of health, chronic kidney disease, polypharmacy and aspirin (100mg).

*Grip strength (kgf) categories: females; normal ≥16 kgf, low <16 kfg. males; normal ≥27 kgf; low <27 kgf

Since the ASPREE population was relatively healthy, targeted interventions such as resistance exercise can assist individuals to aim for higher grip strength [4], we felt that even participants in Q1 can be targeted to aim for a grip strength like Q3−Q4. 

Author Action: We have added the following line in the statistical analysis section (page 7, paragraph 144)

Furthermore, the European Working Group on Sarcopenia in Older People (EWGSOP) [21] was used to examine the association between grip strength and falls.

We have added the following line in the results section (page 11, paragraph 228)

Using the European Cut-Off points for grip strength, we found similar results (S2 Table).

Gait speed: The absolute cut-off point for EWGSOP low gait speed is ≤0.8 m/s for female and male. The ASPREE participants gait speed cut-off points are similar to EWGSOP where female is Q1 ≤0.738 m/s, Q2-4 ≤0.893 m/s and Q5 ≤1.000 m/s. For ASPREE males, Q1 ≤0.814 m/s, Q2-4 ≤0.951 m/s and Q5 ≤1.052 m/s. Thus, reanalysis is not required for gait speed.

2. For Table 2 and 3 – the data were presented with gender-specification but the gender variable was still put into the models.

Author response: We have included a foot note in Table 2 and 3 that the gender variable was not included into the models for gender-specific findings.

Author action: not applicable

3. The results will be more informative if it is segregated by grip strength and gait speed – for example, there could be four matrixes with normal or abnormal grip strength or gait speed – then to evaluate the risk of falls among these four different situations.

Author response: We have performed the analysis as per the reviewer’s suggestion. However, we decided not to present these findings because the risk of falls in the low grip strength, slow gait speed and low grip strength and slow gait speed groups were similar. The HRs and 95% CIs are presented in the table below.

Table: Association between grip strength and gait speed categories and risk of falls in all population (HR, 95% CI)

Grip strength and gait speed categories HR (95% CI)

Neither low grip strength or low gait speed (Q4-5 grip, Q4-5 gait) Reference

Low grip strength only and normal gait speed (Q1 grip, Q4-5 gait) 1.38 (1.19, 1.59)

Low gait speed only and normal grip strength (Q4-5 grip, Q1 gait) 1.46 (1.25, 1.69)

Both low grip strength and low gait speed (Q1 grip, Q1 gait) 1.38 (1.14, 1.67)

Adjusted for age, gender, physical activity, BMI, SF12 State of health, chronic kidney disease, polypharmacy and aspirin (100mg).

The measurement of grip strength is more reliable due to consistency with the dynamometer measurement tools [5]. The limitation of gait speed is the variability of gait speed measurements such as hip joint degeneration [6], fear of falling [7] and requiring at least 4-m walking pathway in a clinic for gait speed measurement [3]. Thus, our study suggests that grip strength and gait speed are equally good at identifying falls and only grip strength can be evaluated to identify people at risk.

Author action: Not applicable.

4. Since this is a longitudinal study – the participants were measured every 6 months or one year – how about using the repeated measure analyses to evaluate the associations.

Author response: In this study we aimed to answer the question if different levels of grip strength and gait speeds are associated with increased risk of falls or not. We did not aim to examine the trajectory of grip strength or the longitudinal grip strength. 

Author action: not applicable

5. This study examines the associations among community-dwelling older population. However, there are more than 20% subjects with polypharmacy and multiple chronic diseases. It would be more informative if the chronic diseases and medications status could be present in the Table 1.

Author response: The ASPREE population were initially healthy and absent of chronic illness expected to limit survival to less than 5 years. Our data shows that 20% of participants were on polypharmacy. Polypharmacy in ASPREE was defined as using ≥3 prescribed medications for any reason. The common reason for the use of medications in our population was to control high blood pressure, high blood sugar or use of statins. For some conditions i.e. high blood pressure, high blood sugar, individuals may take >1 medication. 

We have reported the prevalence of all the risk factors the participants had and if they were on ≥3 medications (polypharmacy).

Author action: not applicable

6. There are some studies using appendicular muscle mass rather than BMI or body fat to examine the association between sarcopenia, frailty and disability among elderly population in recently.

Author response: We do not have the measurements on appendicular mass. Furthermore, for this study, our aim was to understand the association of hand grip strength and gait speed and serious fall-related hospital presentation among elderly population, not to examine the association between appendicular mass and falls.

Author action: not applicable

7. There are some important factors such as the dietary pattern, physical activity and lifestyle could be potential confounders to evaluate the associations.

Author response: We do not have data on dietary pattern, and lifestyle. But we have data on physical activity. In our analyses we have adjusted for physical activity. 

Author action: not applicable

References:

1. Frederiksen H, Hjelmborg J, Mortensen J et al. Age Trajectories of Grip Strength: Cross-Sectional and Longitudinal Data Among 8,342 Danes Aged 46 to 102. Ann. Epidemiol. 2006;16(7):554-562.

2. Dodds RM, Syddal HE, Cooper R et al. Global variation in grip strength: a systematic review and meta-analysis of normative data. Age Ageing. 2016;45(2):209-16

3. Cruz-Jentoft AJ, Bahat G, Bauer J et al. Sarcopenia: revised European consensus on definition and diagnosis. Age Ageing. 2019;48(1):16-31

4. Lee R. The CDC's STEADI Initiative: Promoting Older Adult Health and Independence Through Fall Prevention. Am Fam Physician. 2017;96(4):220-1.

5. Roberts HC, Denison HJ, Martin HJ, Patel HP, Syddall H, Cooper C, Sayer AA. A review of the measurement of grip strength in clinical and epidemiological studies: towards a standardised approach. Age Ageing. 2011 Jul;40(4):423-9. 

6. Tateuchi H, Koyama Y, Tsukagoshi R, Akiyama H, Goto K, So K, Kuroda Y, Ichihashi N. Associations of radiographic degeneration and pain with daily cumulative hip loading in patients with secondary hip osteoarthritis. J Orthop Res. 2016 Nov;34(11):1977-1983. 

7. Asai T, Misu S, Sawa R, Doi T, Yamada M. The association between fear of falling and smoothness of lower trunk oscillation in gait varies according to gait speed in community-dwelling older adults. J Neuroeng Rehabil. 2017 Jan 19;14(1):5.

---

## [Editor Report · Decision Letter 1]

26 Apr 2023

Longitudinal association between handgrip strength, gait speed and risk of serious falls in a community-dwelling older population

PONE-D-23-00361R1

Dear Dr. Hussain,

We’re pleased to inform you that your manuscript has been judged scientifically suitable for publication and will be formally accepted for publication once it meets all outstanding technical requirements.

Kind regards,

Mario Ulises Pérez-Zepeda, M.D., Ph.D.

Academic Editor

PLOS ONE
---

## [Editor Report · Acceptance letter]

29 Apr 2023

PONE-D-23-00361R1 

Longitudinal association between handgrip strength, gait speed and risk of serious falls in a community-dwelling older population 

Dear Dr. Hussain:

I'm pleased to inform you that your manuscript has been deemed suitable for publication in PLOS ONE. Congratulations! Your manuscript is now with our production department. 

Kind regards, 

on behalf of

Dr. Mario Ulises Pérez-Zepeda 

Academic Editor

PLOS ONE